# Host Microbiota Balance in Teenagers with Gum Hypertrophy Concomitant with Acne Vulgaris: Role of Oral Hygiene Associated with Topical Probiotics

**DOI:** 10.3390/microorganisms10071344

**Published:** 2022-07-03

**Authors:** Giovanna Mosaico, Giulia Artuso, Mara Pinna, Gloria Denotti, Germano Orrù, Cinzia Casu

**Affiliations:** 1Independent Researcher, 72100 Brindisi, Italy; 2Department of Surgical Science, Oral Biotechnology Laboratory, University of Cagliari, 09121 Cagliari, Italy; mme.pinna@gmail.com (M.P.); gloriadenotti@gmail.com (G.D.); gerorru@gmail.com (G.O.); 3Independent Researcher, 20123 Milan, Italy; giulia.artuso94@gmail.com

**Keywords:** gingival hypertrophy, acne vulgaris, antibiotic resistance, probiotics, microbioma, microbiota, *Lactobacillus reuteri*, host-microbial symbiosis

## Abstract

Gum hypertrophy is a very frequent condition linked to orthodontic treatment, especially in teenagers, and the same time, about 80% of young adults are affected by acne vulgaris, a chronic inflammatory skin disease, typically treated with antibacterial therapy. The use of probiotics has gained popularity in the medical field, and many studies have demonstrated its effectiveness, such as the positive effects of some bacterial strains belonging to Lactobacillus species. The aim of this study is to document the effect of *Lactobacillus reuteri* (*L. reuteri*) on facial skin that was randomly observed in two orthodontic patients. We present two case reports of a 14-year-old female patient and a 15-year-old male patient suffering from acne vulgaris who, during fixed orthodontic treatment, showed clinical signs of gingivitis with high values of Full Mouth Plaque Score (FMPS) and Bleeding on Probing (BOP). The patients were treated first with professional oral hygiene sessions and Scaling and Root Planing (SRP) procedures, and then with the administration of a formulate containing *L. reuteri* as a probiotic. The follow-up was made at four weeks. During the follow-up analysis, both patients showed a significant clinical remission for gum hypertrophy and skin acne vulgaris.

## 1. Introduction

During fixed orthodontic treatment the amount of plaque is considered an important risk factor that can lead to the development of chronic hyperplastic gingivitis increasing the prevalence of periodontopathogens [1,2]. Several authors have demonstrated that the accumulation of microbial plaque on the teeth surface is strictly connected to gingivitis and periodontitis [3,4,5]. Many teenagers present with gingival infections linked to poor oral hygiene, hormonal changes related to the puberty period and sometimes due to orthodontic treatments, that can cause moderate to advanced periodontal destruction. Orthodontic treatment improves the occlusal and maxillary relationship, the chewing function and also leads to an improvement in the aesthetics of the face; however, harmful effects on the periodontium have been documented.

The presence of fixed appliances and the quantity and quality of the oral microbiota are linked to the control of oral hygiene. Dental braces or retainers placement promotes plaque retention, and consequently, strongly microbial changes in the oral and subgingival microbiota; furthermore, certain specific bacterial species, such as *Aggregatibacter actinomycetemcomitans* (*A. actinomycetemcomitans*), *Porphyromonas gingivalis* (*P. gingivalis*), *Tannerella forsythia* (*T. forsythia*), *Prevotella intermedia* (*P. intermedia*), and *Treponema denticola* (*T. denticola*), which are associated with severe forms of this illness, mainly contribute to developing periodontal diseases [6,7]. Among these main periodontal pathogens which constitute the oral biofilm, the gram-negative anaerobic bacterium *Porphyromonas gingivalis* play a key role in the pathogenesis of periodontal disease and is mostly associated with the chronic form of periodontitis. This non-motile bacterium is an opportunistic oral pathogen that is capable of producing a large number of virulence factors (lipopolysaccharide, capsule, gingipains and fimbriae) and causing gingival inflammation. It is believed that it induces the destruction of periodontal tissues by influencing the host inflammatory response. In fact, the immunological host response is overactivated or blocked by *P. gingivalis* and the other periodontopathogenic species, resulting in temporary or persistent changes and damage to the periodontal tissues that can be seen in histopathological analysis [8]. In this context, SRP treatment performed with the use of manual and ultrasonic instrumentation can be considered the gold-standard therapy for eliminating plaque biofilm. In fact, scaling of teeth surfaces and debridement of root surfaces demonstrated both clinical and microbiological benefits. [9] It is also important to educate the patient with instructions about domiciliary oral hygiene management in order to reduce the inflammatory indices. It is also advisable to act only on plaque deposits trying to avoid damaging soft tissues. In recent years, probiotics showed promising results for the reduction of bacterial load and balancing oral dysbiosis. [10]

Over 80% of teenagers and young adults suffer from acne [11,12]. Acne vulgaris is the eighth most common disease in the world. Acne can be classified into two different types based on the presence of inflammation or not. In the non-inflammatory type, the hair follicle is obstructed but the basement membrane is not damaged; the inflammatory type is instead characterized by papules and/or pustules and we can have damage on the hair follicles and dermal tissues can be observed. *Propionibacterium acnes* (*P. acnes*) can survive and reproduce in deep hypoxic tissue for ≥6 months [13]. Acne vulgaris is clinically characterized by blackheads, inflammatory nodules, pustules and papules on the face, chest and back. In western countries, there is a wider prevalence of this condition; it may be related to diets rich in carbohydrates that increase the insulin-like growth factor signal (IGF-1) [12,14,15].

Some hormones such as dihydrotestosterone and testosterone stimulate the sebum production from sebocytes [16,17]. Interleukin-1 produced by sebaceous glands causes inflammation, hyperkeratinization and formation of comedones. *P. acnes*, also called Cutibacterium acnes, is a gram-positive anaerobic bacterium that belongs to the skin’s normal flora, but in sebaceous glands and hair follicles, it is found in greater quantity [18]. It is also present in the oral cavity, in the conjunctiva, in the bowel and in the external ear canal [19]. It is considered an opportunistic pathogen that can cause a range of infections, including those in the bones, joints and mouth [20]. In the study of Testa M et al., *P. acnes* was found in the oral microbiota composition of a group of forty-five children aged between 6 and 14 (13 boys and 32 girls) [21].

There are many medications proposed for acne treatment, due to the need to find a solution to a chronic disease that often causes scars on the skin [22,23].

Due to the growing problem of resistant bacteria, the restricted use of antibiotics is recommended for both oral and topical routes [24,25].

Tetracycline antibiotics, such as minocycline and doxycycline, are commonly used as the gold standard for the therapy of moderate to severe acne [26]. Recent studies showed that the antibiotic treatment duration often exceeds the recommendations [27,28,29,30]. Many new therapies proposed for acne pathogenesis include sebum-suppressing and anti-inflammatory phytochemicals, laser and light therapy. The research interest in probiotics has increased in the medical field in the last 20 years, especially in their effects on several organs and systems. In the near future, it is possible that therapies for *P. acnes* and acne vulgaris could include probiotics as a gold standard [31].

## 2. Case Report

A 14-year-old female patient and a 15-year-old male patient in systemic good health undergoing treatment with fixed orthodontics, on clinical examination, had numerous sites with bacterial plaque stagnation around the orthodontic brackets, gingival volume enlargement and bleeding on probing (Figure 1 and Figure 2). To improve the clinical picture of gingivitis, a session of professional oral hygiene with SRP was performed with instruction in the correct domiciliary oral hygiene maneuvers (DOH) to be practiced at least twice a day. Specifically, we advised and instructed patients in the use of the rotating-oscillating electric toothbrush with a soft head and in the correct use of spongy dental floss (super floss) equipped with a rigid end to be inserted above or below the orthodontic wire, sliding the spongy part under the gumline of hypertrophic gums. At the end of the DOH, patients were instructed to slowly dissolve a topical probiotic in the mouth. The prescribed topical probiotic was a non-medical commercial formula containing *L. reuteri* ATCC PTA 5289 (American Type Culture Collection, Manassas, VA, USA) in tablets to be taken twice a day, for four weeks (GUM^®^ PerioBalance^®^ Sunstar Italia, Saronno, Italy). Patients were advised not to drink, eat, or rinse their mouths for 60 min after the application. It was also recommended not to rinse the oral cavity with antibacterial mouthwashes in combination with probiotics for the 4 weeks of treatment. Patients were subsequently monitored at 1 week (T1), two (T2) and four weeks (T3). The periodontal clinical parameters were measured and recorded at T0 and re-evaluated at T3 with the online periodontal file “Periodontalchart-online.com” (https://periodontalchart-online.com/uk/, accessed on 30 May 2022). At the four-week follow-up, in both cases there was a marked improvement in hypertrophic gingivitis with a reduction in periodontal epidemiological parameters (Figure 3 and Figure 4). Both patients showed regression of gingival hypertrophy at one month follow-up (100%). The mean plaque index (PI) value before therapy was 69.5%, and at one month it was 1.5%. The mean BoP at T0 was 29% and at T3, one month after treatment, it was 1%. Furthermore, the reduction in inflammation of acne-affected skin was observed surprisingly as both patients also suffered from acne vulgaris. The female patient showed a less dry and brighter skin already after the first week of probiotic therapy; at 2 weeks the improvement was so marked we decided to photograph the skin of the face for an objective evaluation (Figure 5); at T3 a further reduction of acne was observed (Figure 6). Similarly, based on previous results, the male patient received the same probiotic treatment as the female patient (Figure 7). In this second case also, an improvement in skin inflammation was evident; the skin was velvety, hydrated, luminous and almost free of pimples (Figure 8). The patients expressed a high degree of satisfaction with the probiotic treatment, both for gingivitis and especially for acne vulgaris. 

## 3. Discussion

Probiotics are defined as “live microorganisms that, when administered in sufficient concentration, confer a health benefit on the host”. Over the last decade, there has been an increased interest in the use of probiotics especially for periodontal health [32,33]. A recent systematic review showed a positive effect of a combination probiotic of two *L. reuteri* strains as an additive treatment to scaling and root planning [34]. The randomized clinical trials unambiguously showed that in periodontitis patients, this probiotic leads to an improvement of clinical parameters such as a sensible reduction of pocket probing depth after non-surgical mechanical therapy and at the microbiological and immunological level [34,35,36,37]. Vivekananda et al., showed that *L. reuteri* leads to a significant decrease in the CFU (Colony-forming units) of several periodontal pathgens such as *A. actinomycetemcomitans*, *P. gingivalis* and *P. intermedia* [36]. In addition, this probiotic reduced specific parameters associated with inflammation, such as MMP-8 levels in the gingival crevicular fluid [38]. In the pilot study of Sinesi A. et al., 14 adolescent patients with gingival hypertrophy in fixed orthodontic treatment were treated with SRP and topical *L. reuteri*. All the patients showed regression of gingival hypertrophy at the one month follow-up (100%). The initial FMPS value (T0) was 69.71%, and at one month it had fallen to 18.57%. The BoP at T0 was 37.85% and at one month after treatment it was 2.35% [39].

Future research should focus on the underlying immunomodulatory mechanisms of this positive clinical effect of probiotics [40,41].

Human skin microbiota could play an important role in maintaining skin health and preventing premature aging. Although we do not know the actual mechanism of probiotics, some authors have hypothesized that the anti-inflammatory effects are linked to the stimulation of regulatory T cells and the release of IL-10 cytokines. Probiotic strains containing Lactobacillus, Bifidobacterium or Streptococcus and other commensal skin bacteria, have demonstrated cutaneous immunoregulatory effects [42].

Several probiotics can have positive effects on the epidermal tissue. These effects could be connected to the presence of lipoteic acid, which is common in Lactobacillus species [43]. One study highlighted that *Lactobacillus plantarum* (*L. plantarum*) HY7714 is able to prevent UV-induced photoaging by inhibiting MMP-1 expression in dermal fibroblasts [43]. In another study, the authors conclude that oral supplementation of the same type of *L. plantarum* administered for 12 weeks in 110 subjects resulted in greater skin elasticity and hydration [44]. Another study on *Lactobacillus sakei LTA* showed that this microorganism is able to reverse UV-induced skin aging through its immunomodulating effect on monocytes [45]. Topical and oral antibiotics are the first choice in traditional acne treatment protocols. While effective, this approach risks antibiotic resistance and destruction of the microbiome with different systemic collateral events. Given the role of intestinal dysbiosis in inflammatory skin conditions, supplementing probiotics represents a promising alternative or adjuvant approach to treating acne. In a study of 300 patients investigating the administration of *Lactobacillus acidophilus* and *Lactobacillus bulgaricus* for acne, an improvement in acne was observed in 80% of subjects. *Streptococcus salivarius* and *Lactococcus* HY449 both produce an inhibitory substance similar to bacteriocins, which inhibits the growth of *P. acnes* [46,47]. In another clinical study, patients treated with *Lactobacillus* and *Bifidobacterium* species probiotics together with oral antibiotics showed a significantly greater reduction in acne lesion number than in an antibiotic-only control group [48].

We know that probiotics can interfere with the pathogenesis of acne also through immunomodulatory and anti-inflammatory actions. For example, in vitro studies on *Streptococcus salivarius* activity, a commensal microbe, found an important inhibition of IL-8 secretion, suppression of the NF-κB pathway and downregulation of some genes associated with the adhesion of other bacteria to epidermal surfaces.

Probiotics such as *Lactobacillus rhamnosus* SP1 can also reduce the glycemic load, reduce the IGF-1 signaling and finally reduce the proliferation of keratinocytes and hyperplasia of the sebaceous glands [49]. Khmaladze I et al. performed a comparative study on live and lysated products of the probiotic strain *L. reuteri* DSM 17938 in topical skin applications, in which a decrease of IL-6 and IL-8 was observed but live *L. reuteri* DSM 17938 had an antimicrobial action against pathogenic skin bacteria (*Staphylococcus aureus*, *Streptococcus pyogenes M1* etc.), whereas the lysate had no antimicrobial effect. Therefore, it is hypothesized that *L. reuteri* DSM 17938 may be useful for general skin health and for improving the skin barrier [50]. Kang MS et al. successfully examined the inhibitory effects of *L. reuteri* on the proliferation of *P. acnes* and *S. epidermidis*, which are closely linked to the production of organic acids. Overall, these results suggest that *L. reuteri* may be a useful probiotic agent for controlling the growth of bacteria involved in skin and gum inflammation [38,51].

All in vitro studies have shown that probiotics successfully inhibited skin and wound pathogens and promoted healing. The exogenous and oral application of probiotics has shown a reduction in infections, especially if used as an adjuvant to antibiotic therapy, and therefore, the use of probiotics in this field remains worthy of further studies [52].

## 4. Conclusions

These two cases suggest that *L. reuteri* had a positive effect on the growth control of bacteria involved in acne vulgaris in teenagers and gingival plaque-dependent hypertrophic gingivitis. The results also suggest that professional oral hygiene treatment combined with the topical probiotic *L. reuteri*, could benefit the microbiota eubiosis status, which manifests itself, in particular, on the cutaneous and oral microbiota with a reduction of inflammation and swelling of the gums and pimples; it also prevents the worsening of these two conditions. These cases highlight a possible correlation in the etiopathogenesis of these two conditions.

This preliminary work could be a useful base protocol for the management of patients simultaneously affected by gum hypertrophy and refractory skin acne vulgaris. In this context, a multidisciplinary approach based on various medical fields remains important to promote the microbial symbiosis of the host. No firm conclusion can be drawn from a such small sample but our study offers interesting outcomes for further research.

## Figures and Tables

**Figure 1 microorganisms-10-01344-f001:**
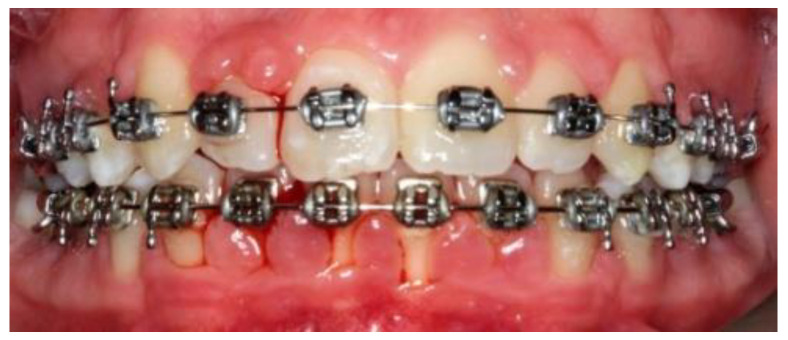
Patient n.1, T0: particular gingival swelling, accompanied by bleeding and abundant accumulation of bacterial plaque.

**Figure 2 microorganisms-10-01344-f002:**
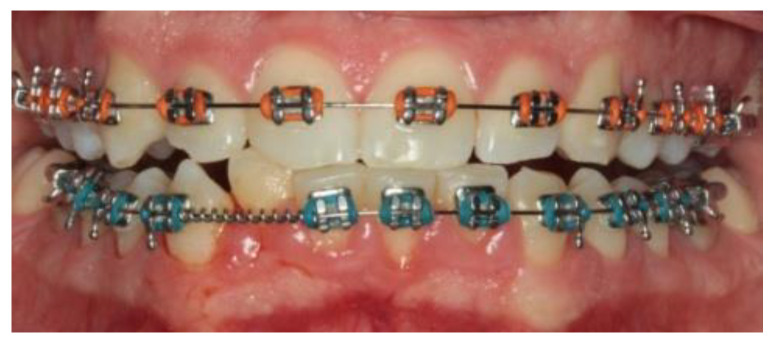
Patient n.2, T0: gingival hypertrophy localized to the lower incisors.

**Figure 3 microorganisms-10-01344-f003:**
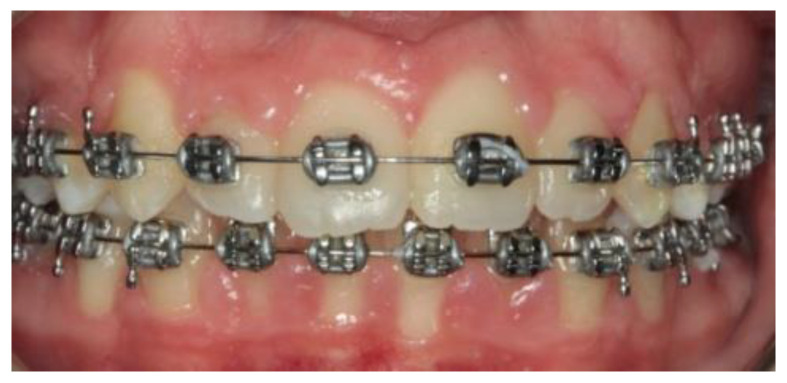
Patient n.1 control at 4 weeks (T3): zero bleeding, reduction of periodontal indices.

**Figure 4 microorganisms-10-01344-f004:**
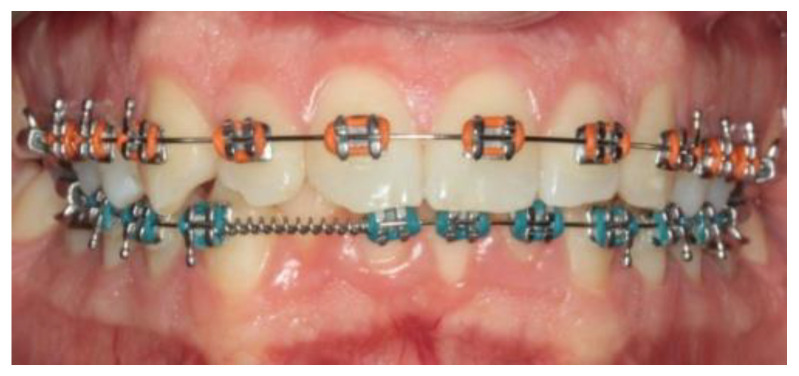
Patient n.2, T3: at the 4-week check-up, the gums of the fifth sextant were healed.

**Figure 5 microorganisms-10-01344-f005:**
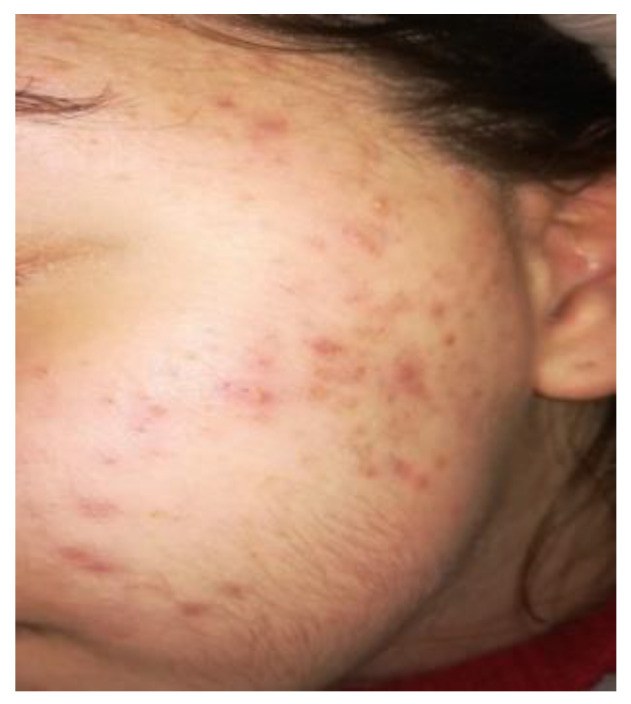
Two weeks follow up, patient n.1, T2: patient 2 weeks after the start of therapy with *L. reuteri*.

**Figure 6 microorganisms-10-01344-f006:**
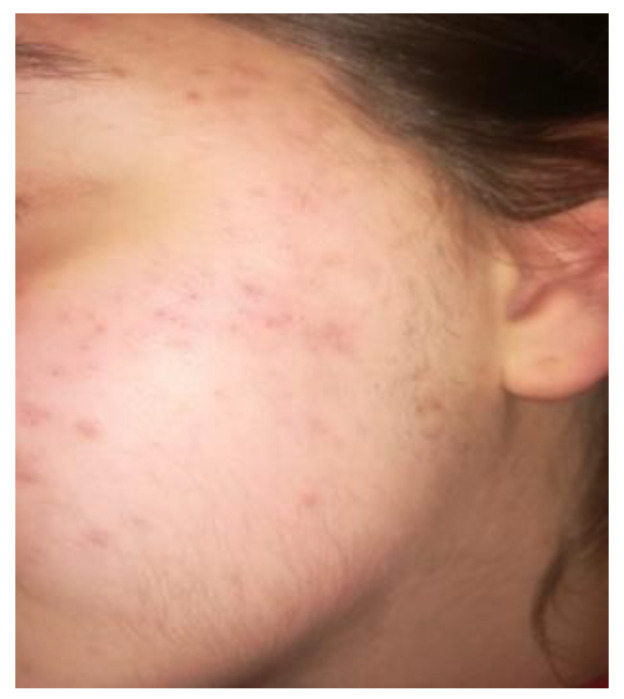
T3 patient n.1: picture of the patient’s skin with acne reduction four weeks after therapy with *L. reuteri*.

**Figure 7 microorganisms-10-01344-f007:**
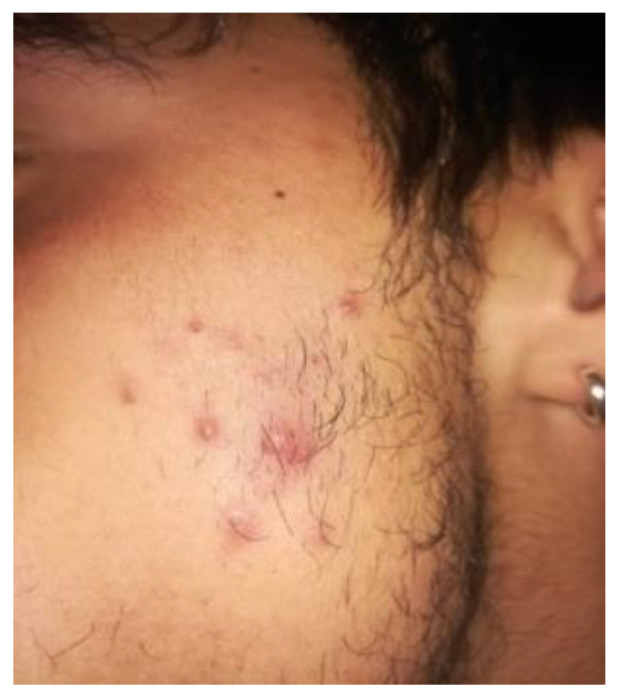
Patient n. 2 T0: acne with pustules before treatment with *L. reuteri*.

**Figure 8 microorganisms-10-01344-f008:**
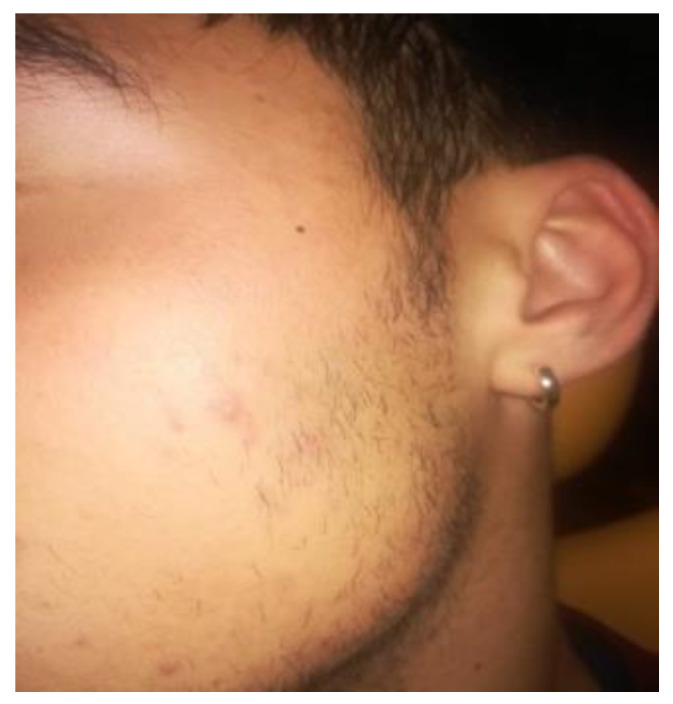
T3 patient n.2: almost complete cure of acne after 4 weeks of treatment with *L. reuteri*.

## Data Availability

Not applicable.

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
