# Peer review of "Host Microbiota Balance in Teenagers with Gum Hypertrophy Concomitant with Acne Vulgaris: Role of Oral Hygiene Associated with Topical Probiotics"

_microorganisms, 2022, doi:10.3390/microorganisms10071344_

Round 1
Reviewer 1 Report
An interesting observation but unfortunately you need some editing .
Remember that not objective conclusion or recommendations can be made from 2 clinical cases. But the observation is interesting and needs further studies in a case control study designed.
Author Response
Dear Reviewer,
We all thank your for review and suggestions on our document. We reviewed the manuscript as you suggested. Here you can find the corrections we have made. You can also find them in red in the latest version of the newspaper.
1) Thanks for your suggestion. It was immediately corrected.
2) Thanks for the comment, we have proceeded to rework the conclusions as recommended by you. I hope it gets better now.

Reviewer 2 Report
The article focused on “Host microbiota balance in teenagers with gum hypertrophy concomitant with acne vulgaris” The article is very interesting. There were some suggestions:
1. In Abstract. Line 12. Suggest to provide the full name of “SRP” when first appeared.
2. In Abstract. Since it is a case report article, suggest to focus on the “case” instead of listing the background. Suggest to revise the Abstract part.
3. In Introduction part. Line 30. It is a little strange to say “etiological factor”. Suggest “etiology” or “risk factor”
4. In Introduction part. Line 31. “Several authors….” But only one citation.
5. It is a case report related to gingivitis. But most of the Introduction, line 37-74, focused on facial acne. Suggest revise the Introduction part to have more introduction on “gingivitis, microbiota and probiotics”
6. For line 75-106. The section should be “case report” instead of “material and method” and “result”. Moreover since it is a case report, suggest have more detail description of the case.
7. Line 81 “After signing the parental consent form…” Why sign this form? Is it a clinical trial? If yes, please provide the IRB number.
8. In Discussion part. Line 121-123 “At the first….. systemic effects” This section is not direct related to the Results and is unnecessary.
9. The article need English editing.
Author Response
Dear Reviewer,
We all thank you for your revision and your suggestions on our paper. We reviewed the manuscript point by point, as you suggested to us. Here, you can find the corrections we made. You can also find them in red last version of the paper.
1) Thank you for your suggestions. It has been immediately corrected.
2) Thank you for your kind suggestion. We review the whole abstract in order to focalize on the "case reports".
3)4)5) Thank you for your corrections. We processed again this entire part in order to focus on the actual theme, as you suggested.
6) Thank you for your piece of advice. We changed the name section and added details of the cases.
7) Thank you for your valuable suggestion. We confirm that our work does not include a clinical trial; parents have signed informed consent for a general treatment that included a topical commercial probiotic product. However, we have removed on the text this sentence to avoid any misunderstanding.
8) Thank you for your suggestion. We promptly reviewed this part.
9) Thank you for your advice. We reviewed the entire manuscript; we hope you may find last version written in a better English.

Round 2
Reviewer 2 Report
had been revised as suggestions